# Major Hepatectomy En Bloc with Cava Vein Resection for Locally Invasive Caudate Lobe Hepatocarcinoma

**DOI:** 10.3390/healthcare9101396

**Published:** 2021-10-19

**Authors:** Nicolae Bacalbasa, Irina Balescu, Florin Ichim, Ion Barbu, Alexandru Ristea, Razvan Lazea, Ioana Danciuc, Ioana Popa, Ovidiu Magdoiu, Gabriela Smira, Camelia Diaconu, Florentina Furtunescu, Ovidiu Stiru, Cornel Savu, Claudia Stoica, Vladislav Brasoveanu, Bogdan Ursut, Adnan Al Aloul

**Affiliations:** 1Department of Visceral Surgery, Center of Excellence in Translational Medicine “Fundeni” Clinical Institute, 022328 Bucharest, Romania; florinichim@yahoo.ro (F.I.); ionbarbu@yahoo.ro (I.B.); alexandruristea@yahoo.ro (A.R.); razvanlazea@yahoo.ro (R.L.); ioanadanciuc@yahoo.ro (I.D.); ioanapopa@yahoo.ro (I.P.); ovidiumagdoiu@yahoo.ro (O.M.); vladislavbrasoveanu@yahoo.ro (V.B.); 2Department of Obstetrics and Gynecology, Carol Davila University of Medicine and Pharmacy, 050474 Bucharest, Romania; 3Department of Surgery, “Ponderas” Academic Hospital, 014142 Bucharest, Romania; irina_balescu206@yahoo.com; 4Department of Gastroenterology, “Fundeni” Clinical Institute, 022328 Bucharest, Romania; gabrielasmira@yahoo.ro; 5Department of Internal Medicine, “Carol Davila” University of Medicine and Pharmacy, 050474 Bucharest, Romania; cameliadiaconu@yahoo.ro; 6Department of Internal Medicine, Clinical Emergency Hospital of Bucharest, 014461 Bucharest, Romania; 7Department of Public Health and Management, University of Medicine and Pharmacy “Carol Davila”, 050474 Bucharest, Romania; florentinafurtunescu@yahoo.ro; 8Emergency Institute for Cardiovascular Diseases, 022328 Bucharest, Romania; ovidiustiru@yahoo.ro; 9Department of Cardio-Thoracic Pathology, “Carol Davila” University of Medicine and Pharmacy, 050474 Bucharest, Romania; 10Department of Thoracic Surgery, “Marius Nasta” National Institute of Pneumology, 050159 Bucharest, Romania; cornelsavu@yahoo.ro; 11Department of Thoracic Surgery, “Carol Davila” University of Medicine and Pharmacy, 050474 Bucharest, Romania; 12Department of Surgery, Ilfov County Hospital, 077160 Bucharest, Romania; claudiastoica@yahoo.ro; 13Department of Anatomy, Carol Davila University of Medicine and Pharmacy, 050474 Bucharest, Romania; bogdanursut@yahoo.ro; 14Department of Surgery, Emergency Hospital “Agrippa Ionescu”, 011356 Bucharest, Romania; 15Department of Surgery, Ramnicu Sarat County Hospital, 125300 Buzau, Romania; adnanalaloul@yahoo.com; 16Facullty of Medicine, “Titu Maiorescu” University of Medicine and Pharmacy, 031593 Bucharest, Romania

**Keywords:** caudate lobe, hepatocarcinoma, major hepatectomy, cava vein resection

## Abstract

Background/Aim: Locally advanced liver tumours with vascular invasion have been considered for a long period of time as unresectable lesions, so the patient was further deferred to oncology services for palliation. However, improvement of the surgical techniques and the results reported so far came to demonstrate that extended hepatic and vascular resections might be safely performed in such cases and might significantly improve the long-term outcomes. Materials and Methods: A 61-year-old patient was diagnosed with a caudate lobe tumour invading the inferior cava vein and the right hepatic pedicle. Results: The patient was successfully submitted to surgery, and an extended right hepatectomy en bloc with cava vein resection was performed; the continuity of the cava vein was re-established by the placement of a synthetic graft. The postoperative outcome was uneventful. Conclusions: Although initially considered as a formal contraindication for resection, vascular invasion of the greater vessels should not preclude surgery if complete resection is achievable.

## 1. Introduction

Locally advanced hepatocellular carcinoma has been considered for a long period of time as not suitable for surgery, especially in cases in which invasion of the surrounding vascular structures is reported. In such cases, liver transplantations were initially proposed, but the long-term outcomes reported unacceptable rates of local recurrence and therefore, these patients were further excluded from the waiting lists. However, recent studies came to demonstrate that in such cases, radical surgery might improve the long-term outcomes of these cases [1]. The aim of the current paper is to report the case of a male patient in which right hepatectomy en bloc with the caudate lobe and the inferior cava vein was successfully performed.

## 2. Case Report

After obtaining the approval of the Ethical Committee, the data of the patient were retrospectively reviewed and presented in the current paper. The 61-year-old patient with no significant medical history was investigated for weight loss of eight kilograms in the last two months, and asthenia and was diagnosed at the abdominal ultrasound with a large tumour located at the level of the caudate lobe. The patient was further submitted to a computed tomography which confirmed the presence of a unique lesion at the level of the caudate lobe, measuring 10.67/6.01/8.46 cm and invading the inferior cava vein as well as the right hepatic pedicle (Figure 1, Figure 2 and Figure 3); meanwhile, the remaining liver parenchyma presented no signs of cirrhotic transformation and no other tumoral lesions. 

Meanwhile, the blood test results revealed the presence of increased serum values of alpha fetoprotein [over 1000 ng/mL] while the viral markers for B, C and D hepatitis were negative; serum levels of carcinoembryonic antigen were slightly increased (15 ng/mL). In order to exclude the metastatic origin of the lesions, the patient was submitted to an upper and, respectively, lower digestive endoscopy which demonstrated the absence of other tumoral lesions. Furthermore, the patient was submitted to liver biopsy which demonstrated the presence of moderately differentiated hepatocarcinoma (Edmonson Steiner II/III lesion). After discussing the risks and benefits with the patient, he was further submitted to right hepatectomy extended to the caudate lobe en bloc with segmental resection of the inferior cava vein. The continuity of the inferior cava vein was re-established by placing a synthetic Gore-Tex graft, while the patency of the left hepatic pedicle was checked by intraoperative ultrasound (Figure 3, Figure 4 and Figure 5). 

The postoperative outcome was a simple one, with the patient being discharged on the 10th postoperative day; the computed tomography which was performed before discharge revealed the presence of a functional venous graft as well as a normal aspect of the remaining liver parenchyma (Figure 6). 

The histopathological study confirmed the presence of a 7.5/9/8.5 poorly differentiated hepatocarcinoma in association with focal vascular invasion of the inferior cava vein; meanwhile, the resected specimen of the cava vein presented a total length of 3 cm and presented an intraluminal tumour thrombus. The immunohistochemical studies demonstrated the presence of Hyaluronic Acid Synthases (HAS) and Arginase 1 (Arg 1) in association with the absence of cytokeratin (CK) 7, CK20 and, respectively, CK 19, demonstrating therefore the presence of a hepatocellular carcinoma. Meanwhile, the non-tumour liver parenchyma proved to have no particular histopathological aspect while the specimen presented negative resection margins. The postoperative course was favourable; at six months follow-up, the serum levels of AFP and CEA were of 12 UI/mL and 5 ng/mL, respectively, while the imagistic studies failed to demonstrate any sign of local or distant recurrence.

## 3. Discussion

Local invasion of the greater vessels in patients with liver tumours has been considered for a long period of time as unjustified, and therefore, such cases have been submitted to palliative oncological procedures [1,2,3,4,5]. In such cases, certain authors proposed liver transplantation; however, the long-term outcomes reported poor survival results. This fact, in association with the global paucity of the liver graft, led to the exclusion of these cases from the waiting lists for liver transplantation [6]. However, the improvement of vascular surgical techniques of resection and reconstruction, the introduction of novel procedures such as ex vivo liver resection or total vascular exclusion, as well as the improvement of perioperative management led to the opportunity of performing extended hepatectomies in association with major vascular resections [6,7,8,9,10,11,12,13].

Due to the rarity of cases submitted to such complex procedures, certain authors conducted studies in which they included patients with different origins of the primary tumours, the common element in these cases being represented by the necessity of cava vein resection. One of the largest studies has been recently published by Ruiz et al. in the Journal of Vascular Surgery; the study included 52 patients submitted to caval resections for different malignancies, graft placement being needed in 33% of cases. The authors reported an overall survival at the two-year follow-up of 64.7%; the most commonly reported indication was represented by cava vein sarcoma (in 15 cases), while hepatocellular carcinoma was reported in a single case. However, the authors reported that hepatic resections were needed in 12 cases. As for the type of graft, the authors reported that autologous grafts were used in only three cases, while in the remaining cases, synthetic grafts were placed [14].

An interesting study which analysed the efficacy and safety of major vascular resection for locally advanced liver malignancies was conducted in 2017. The study was a literature review of 38 papers; among the patients included in this paper, major morbidities were reported in 43% of cases, the most commonly encountered one being related to the development of liver failure. As for the postoperative mortality, a rate of 5% was reported, the most common cause of death being again represented by liver failure (in seven cases) followed by septic shock in five cases, transfusion related graft versus host disease and respectively pulmonary haemorrhage—each in one case. When it comes to the type of liver resection, as expected, the most complex one was represented by extended right hepatectomy, reported in 21% of cases. As for the type of cava vein resection, circumferential resection followed by synthetic graft replacement was the option of choice in 107 cases. As expected, most surgical procedures were performed after total vascular exclusion (64% of cases) followed by veno-venous bypass in 29% of cases, and ante situm resection in and ex situ resection each in 4% of cases. When analysing the histopathological reports, vascular invasion was found in 38% of cases, the remaining patients presenting only fibrous adhesions between the inferior cava vein and the tumour; however, the authors underlined the fact that, in cases in which the preoperative imagistic studies cannot precisely distinguish between tumoral invasion and fibrosis, en bloc resection would be performed in order to avoid the development of haemorrhagic incidents and, in the meantime, prevent the possible dissemination of the tumoral cells when manipulating the tumour. As for the long-term outcomes, the authors reported a 5-year overall survival rate of 33%, being influenced by the type of pathology which imposed performing such extended liver procedures; therefore, the 5-year survival rate was of 26% for cases submitted to surgery for colorectal liver metastases, 37% for cases with intrahepatic cholangiocarcinoma and 30% for hepatocellular carcinoma; interestingly, the long-term outcomes were not significantly influenced by the presence of the invasion of the venous wall. Therefore, the authors concluded that local vascular invasion of the cava vein is rather related to the location of the tumour in close proximity of this vascular structure and not by a more aggressive biology of the tumour. When analysing the type of reconstruction, the authors underlined the fact that graft placement should be the option of choice in cases in which the length of the resected venous segment is larger than 2 cm, autologous or synthetic grafts being proposed. Although autologous grafts should be preferred due to the lower infectious and thrombogenic risks, these might not be always available, especially if a longer segment is resected; in such cases, synthetic grafts should be used [6]. However, it seems that the infectious risk is higher whenever biliary or bowel resection is associated, with autologous reconstruction being favoured in such cases [15]. Another strategy which has been proposed in highly selected cases is represented by caval ligation without reconstruction; this surgical approach is reserved for cases presenting a patent collateral venous circulation which is able to maintain an adequate venous return. As expected, cases which are suitable for this surgical approach are represented by those presenting complete caval obstruction which was developed during a long period of time [15,16]. However, in such cases, serious postoperative complications such as renal failure or extended deep venous thrombosis might occur [16,17].

As for the postoperative risks in cases in which synthetic grafts are used, it seems that the thrombotic and infectious complications are the most commonly encountered; however, in cases in which graft thrombosis occurs, minimal clinical impact is expected due to the common presence of a significant collateral network [14]. When it comes to the infectious risk, it seems that it is significantly increased by the association of concurrent digestive tract resections and it can be prevented by omental placement; however, in such cases, autologous arterial or venous grafts should be also taken in consideration due to the fact that, while unlikely for thrombotic complications, the infectious ones are considered to be life-threatening ones [16,18,19,20]. Therefore, in cases in which caval resection is dictated by the presence of a locally advanced hepatocellular carcinoma in the absence of any digestive resections, synthetic grafts are expected to be well tolerated. Moreover, the long-term outcomes demonstrated similar rates of survival for cases presenting microscopic invasion of the venous wall when compared to those in which invasion could not be demonstrated by the histopathology report. This fact enables us to consider that vascular invasion in such cases is rather related to the location of the primary tumour and not to the biological behaviour of the neoplastic lesion.

As for the indications of palliative treatment, in certain cases in which multiple tumoral lesions are found in both the hepatic lobes as well as in cases in which portal vein invasion is encountered, palliative treatment is recommended. However, in such cases, the overall prognostic remains extremely poor, the overall survival ranging between several months and one year and a half [20,21].

## 4. Conclusions

Although rarely performed, extended hepatic resections in association with cava vein resection have been successfully reported so far in selected cases diagnosed with locally advanced liver lesions. The type of venous reconstruction can widely vary depending on the local anatomical conditions and range from simple ligation of the vein followed by no reconstruction (in cases presenting an adequate collateral network) to autologous graft placement (in cases in which the length of the resected segment allows it) and synthetic graft placement (in cases in which more extended resections are needed). Although synthetic graft placement is associated with higher postoperative risks such as thrombosis or infection, we should not omit the fact that most often, thrombosis is well tolerated, especially due to the presence of patent collateral venous vessels, while infection is more commonly expected whenever digestive resections are also associated.

## Figures and Tables

**Figure 1 healthcare-09-01396-f001:**
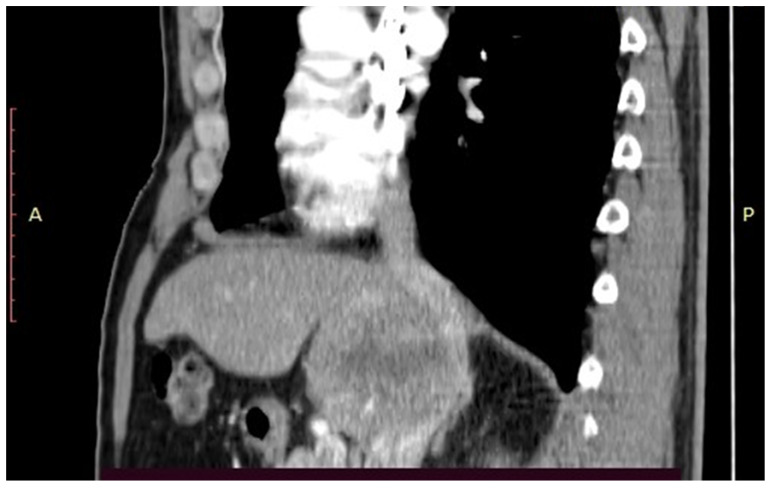
Computed tomography demonstrated the presence of a large tumour invading the inferior cava vein.

**Figure 2 healthcare-09-01396-f002:**
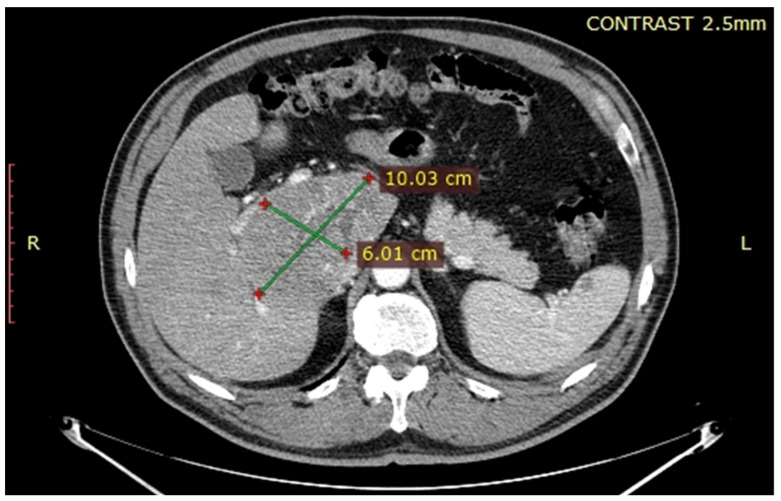
Computed tomography also revealed the presence of local invasion of the right hepatic pedicle.

**Figure 3 healthcare-09-01396-f003:**
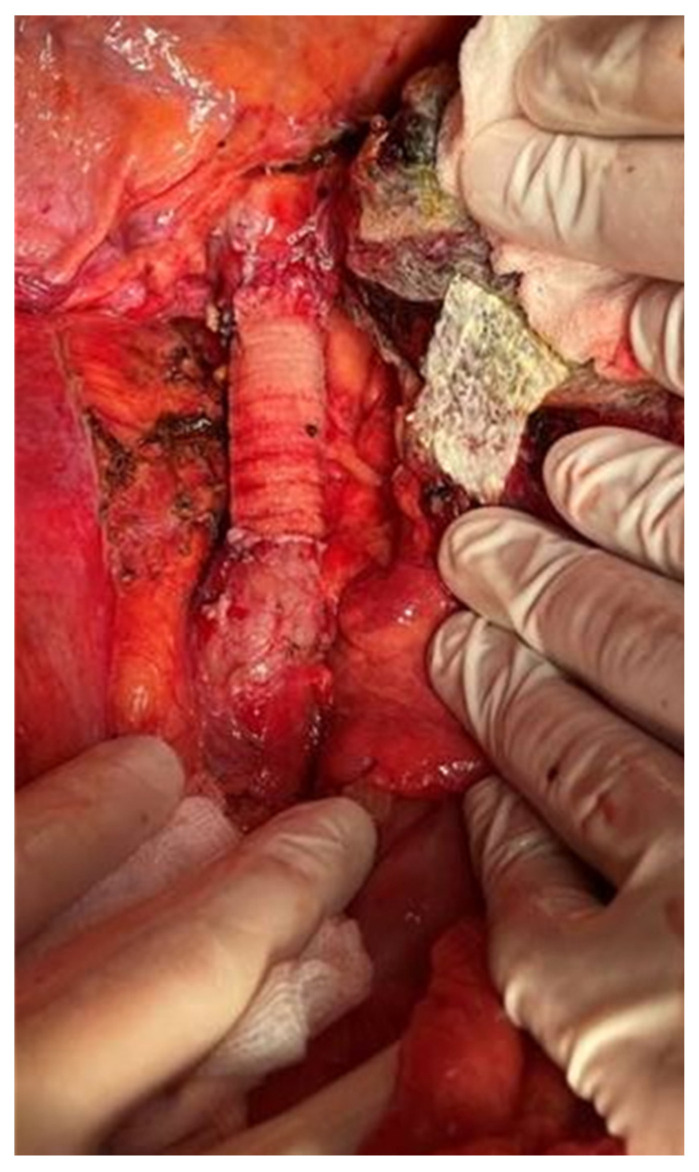
The final aspect after resection: the continuity of the cava vein was re-established by placing a Gore-Tex graft; the left suprahepatic vein and its implantation was successfully preserved.

**Figure 4 healthcare-09-01396-f004:**
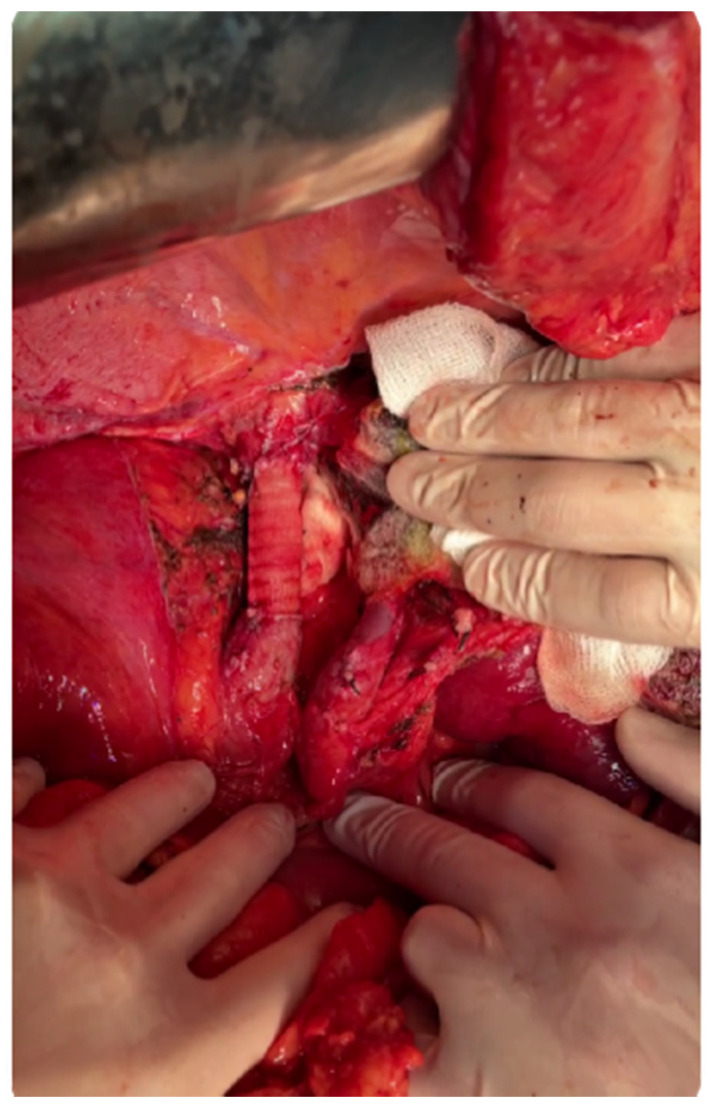
Intraoperative aspect revealing the presence of the reconstructed cava vein by Gore-Tex graft placement, as well as the remaining left hepatic pedicle (left portal vein, left hepatic artery, left bile duct).

**Figure 5 healthcare-09-01396-f005:**
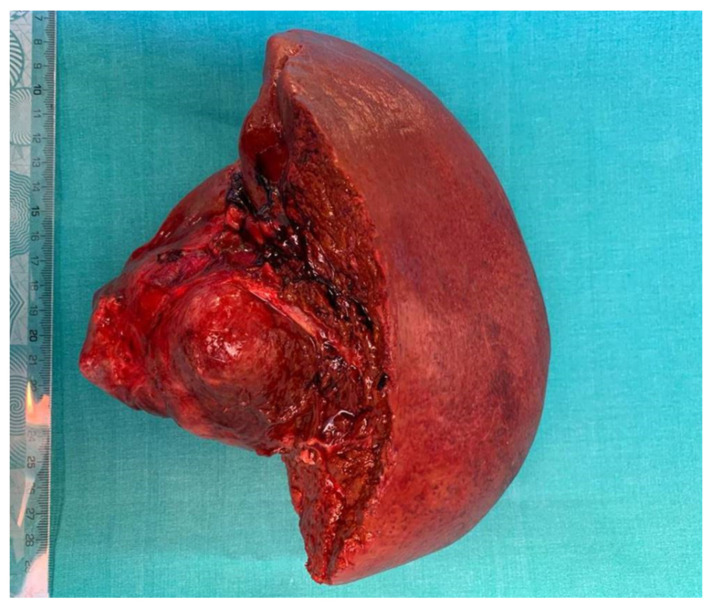
The specimen of right hepatectomy extended to the caudate lobe and to inferior cava vein with a total weight of 1100 g, measuring 22/16/10 cm.

**Figure 6 healthcare-09-01396-f006:**
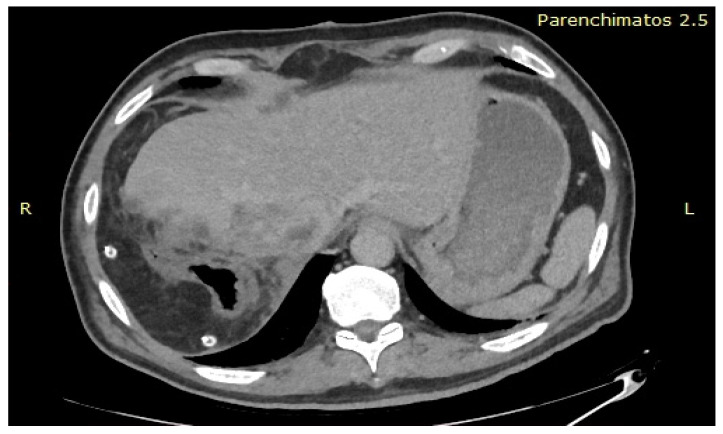
Computed tomography revealing the patency of the graft as well as of the left hepatic pedicle.

## Data Availability

Data supporting the reported results are available from the corresponding author at request.

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
