# Peer review of "Major Hepatectomy En Bloc with Cava Vein Resection for Locally Invasive Caudate Lobe Hepatocarcinoma"

_healthcare, 2021, doi:10.3390/healthcare9101396_

Reviewer 1 Report

Authors reported a case report with the aim of reporting successful right hepatectomy en bloc with the caudate lobe with the inferior cava vein resection and graft placement as palliative oncological procedures. The artice is well illustrated and referenced. The description of the case is given thoroughly. Although this is a clear case report, I would like to propose just one issue to discuss.

1. It would be great to have a clear and comprehensive description of strategies which should be used as palliative operation methods in association with venous reconstruction during extending hepatic resections. Please, make the conclusive part much more legible and pervasive

Author Response

Thank you for taking your time to review our paper. We modified the text according to your demands; 

Reviewer 2 Report

In their manuscript the authors present a case report of of a male patient who successfully underwent right hepatectomy en bloc with the caudate lobe and the inferior cava vein. I would like to commend the authors/surgeons for performing what seems as indeed a challenging procedure yet their manuscript work is not as novel.  

  • R status, readmission status, 90-day mortality or long-term outcomes are not provided.
  • Indeed the literature was somewhat poor in presentation of complex resections 15 years ago. However now multiple series have been published.
  • The quality of language could be partly improved.

Author Response

(The authors gave the same response as above.)

Reviewer 3 Report

This case report described a patient with HCC involved in caudate lobe and received right lobectomy and caudate lobe resection including invovled IVC.

Comments

  1. Several key data are missing, including liver panels (before and after operation), long-term changes of AFP, F/U images (at least 1 month post operation), and F/U period.
  2. Please check the language. In abstract result section:    the patient was successfully submitted to surgery, an extended right hepatectomy en bloc with cava vein resection was performed; due to the extended length of the resected cava vein reconstruction with synthetic graft was performed.  "due to" here is a bit strange. Similarly, in line 138-142, the sentence is too long and I would suggest it to break into several sentences. 
  3. Line 109, "local vascular invasion", do you mean IVC or other intreahepatic vessel?  line 111, "intramural thrombus", is it a tumor thrombus?
  4. Line 114, "primitive hepatocarcinoma", can you define it more clear? Do you mean hepatocellular carcinoma? what do you mean primitive?
  5. How about the non-tumor liver look like in pathology? Do this patient have any chronic liver disease anyway? NASH or alcoholic liver? 

Author Response

Thank you for taking your time to review our paper. We modified the text according to your demands; unfortunately we don’t have any follow up images except the one of the computed tomography which was performed before discharge.

Round 2

Reviewer 2 Report

The authors have addressed my remarks.

Author Response

Thank you for your time

Reviewer 3 Report

I have no other comments.

Author Response

Thank you for your time and response!
